# Audio Image Generation for Denoising

## Abstract

Time-frequency domain analysis has emerged as an effective method to remove noise in audio signals. However, the image generation quality of the frequency domain is not yet well explored. In this paper, we turn the audio denoising task into an image generation problem. We present an audio image generation model for audio denoising named AIGD and use it to estimate the posterior distribution of clean complex images conditioned on noisy complex images. Given any noisy audio signals, our AIGD model could directly generate denoised complex images and output clean audio signals. We further optimize complex L2 and complex absolute structure similarity losses to improve the quality of generated images. An SDR loss is proposed to reconstruct better-denoised audios. Extensive experimental results demonstrate that by generating high-quality frequency domain images, our AIGD model achieves state-of-the-art performance audio denoising.

## 1 Introduction

Audio signals are commonly used as a means of information transmissions, such as cellular communication and teleconferences (Kong et al., 2021), human-robot communication (Xu et al., 2020). Other than communication, audio signals are also used for disease diagnoses, e.g., lung sound (Pouyani et al., 2022) and heart sound (Kui et al., 2021) detection, and other practical applications. Different from human-human communication, these applications depend greatly on the quality of audio signals. In a real-life environment, audio signals are inevitably affected by natural and man-made noises, resulting in lower-quality audio signals with reduced transferable information. To cope with this challenge, audio denoising becomes an important technique to remove unwanted noise in the audio to retain as much useful information as possible.

The key to audio denoising is to separate clean audio signals from noises to form high-quality transferable signals. Some traditional methods were developed to perform this task, such as spectral subtraction (Boll, 1979; Berouti et al., 1979), Wiener filtering (Chen et al., 2006; Lim & Oppenheim, 1979), and Wavelet transformation (Zhao & Cui, 2015; Srivastava et al., 2016). However, the performance of these methods is constrained by the presence of natural noise. The introduction of deep learning methods has revolutionized the audio-denoising field with a stronger ability to learn data and characteristics with fewer samples. Time-frequency (TF) domain deep-learning-based audio denoising methods (Sonning et al., 2020; Wang et al., 2021), waveform domain audio denoising methods (Kong et al., 2022), and CNN/RNN-based audio denoising methods (Abouzid et al., 2019; Alamdari et al., 2021) are explored and demonstrated better performance in denoising tasks.

However, these models also encounter challenges. One common challenge is that audio signals that come with different natural noises are difficult to be separated to extract clean audio signals. Diffusion probabilistic models (Lu et al., 2021) try to counter this challenge by adding Gaussian noise to clean signals and using a reverse process to sample the distribution of the original audio and reconstruct clean signals. However, this approach starts with clean audio signals, while in a real-world application, noisy audio signals are usually only provided. In addition, adding noise to clean audio signals is an unnecessary process, given noisy audio exists.

To alleviate the aforementioned challenges, our contributions are three-fold:

- We convert the audio denoising into an image generation task. Experimental results demonstrate that better-generated complex images achieve higher audio-denoising performance.

- We remove the forward process in the diffusion model and develop an image generation process in which we generate high-quality complex images to estimate the posterior distribution of clean complex images conditioned on noisy complex images.

- We also propose image quality check and audio reconstruction modules. We enforce complex L2 loss and complex absolute structural similarity loss for the image quality check and employ an SDR loss to optimize the audio reconstruction module.

## 2 RELATED WORK

**Time-frequency Models.** Deep learning methods in the audio denoising field demonstrate a stronger ability to learn data features. In recent years, time-frequency (TF) domain approaches that work on the spectrogram become mainstream. Hu et al. (2020) proposed a deep complex convolution recurrent network (DCCRN) model that combined deep complex U-net and convolution recurrent network, using long short-term memory (LSTM) to model temporal context with reduced trainable parameters and computational cost. Lv et al. (2022) extended the previous DCCRN model to a super wide band version and employed a complex feature encoder (CFE) after short-time Fourier transform (STFT), and a complex feature decoder (CFD) before inverse short-time Fourier transform (ISTFT). Sonning et al. (2020) investigated the performance of a time-domain network. The model was developed to deal with the original inability of STFT/ISTFT-based time-frequency approaches to capture short-time changes and was proved to be useful in a real-time setting.

**Recurrent Neural Networks & Waveform Domain.** The aforementioned methods have a similar feature of using clean audio signals as training targets, meaning that their performance will be limited in low signal-to-noise ratio (SNR) scenarios (Zhou et al., 2023). To deal with this problem, enhanced deep learning approaches are proposed to better classify audio signals. Zhang et al. (2016) built a novel deep recurrent convolutional network for acoustic modeling and applied deep residual learning for audio recognition with faster convergence speed. Chen & Wang (2017) proposed an RNN-based audio separation model with four hidden LSTM layers to deal with unseen speakers and unseen noises regarding objective speech intelligibility. Tan & Wang (2018) proposed a recurrent convolutional network that incorporated a convolutional encoder-decoder and LSTM into the convolutional recurrent neural network (CRN) architecture to address real-time audio enhancement. Li et al. (2020) combined the progressive learning framework with a causal CRN to further mitigate the trainable parameters and improve audio quality and intelligibility. Audio denoising in the waveform domain has also been explored. Defossez et al. (2020) presented a causal speech enhancement model based on an encoder-decoder architecture with skip connections. It optimized both time and frequency domains and was capable of removing both stationary and non-stationary noises. Kong et al. (2021) proposed an audio enhancement method with pre-trained audio neural networks (PANNs) and applied a convolutional U-Net to predict the waveform of individual anchor segments selected by PANNs. Kong et al. (2022) proposed a causal speech denoising model, CleanUNet, on the raw waveform based on an encoder-decoder architecture combined with self-attention blocks. The model was optimized through a set of losses defined over both waveform and multi-resolution spectrograms.

**Diffusion Models.** The diffusion probabilistic model (Sohl-Dickstein et al., 2015), is also introduced to deal with audio denoising. The model has demonstrated great capacity in generating high-quality images (Ho et al., 2020; Nichol & Dhariwal, 2021) and raw audio waveform (Kong et al., 2020; Liu et al., 2021). The diffusion probabilistic model includes a forward diffusion and a reverse process. The forward process converts clean input data to an isotropic Gaussian distribution by adding Gaussian noise to the original signal at each step. In the reverse process, the model subtracts the predicted noise signal from the noisy input to retrieve the clean signal. Lu et al. (2021) proposed a diffusion probabilistic model-based speech enhancement (DiffSE) model based on the DiffWave model (Kong et al., 2020). Lu et al. (2022) proposed an enhanced conditional diffusion probabilistic model that can adapt to non-Gaussian real noises in the estimated speech signal in its reverse process. Chen et al. (2023) designed an actor-critic-based framework to guide the reverse process to the metric-increasing direction. However, these diffusion-based models add noise to clean images in the forward process and then denoise newly made noisy images in the reverse process, which violates the rule of audio denoising. In real-world scenarios, only noisy audio signals are provided. In this paper, we directly denoise noisy audio signals using an image generation process.

## 3 METHOD

### 3.1 MOTIVATION

Although audio denoising in the TF domain has been well explored, the image generation quality of the frequency domain is not addressed. In addition, existing diffusion models first added noises to clean audio. Then, they performed an audio denoising task, which violates the original datasets given noisy audios already existed, and the newly generated noisy audios are different from the original noisy audios. We aim to remove the unnecessary forward process of diffusion models and develop a novel image generation process to pursue high-quality generated images, which is equivalent to obtaining high-quality denoised audio signals.

### 3.2 PROBLEM

In time domain audio denoising, a noisy audio signal $x$ can be typically expressed as:

$$x = y + noise \tag{1}$$

where $y$ and $noise$ denote clean audio and additive noise signal, respectively. Given noisy audio signals $X = \{x_i\}_{i=1}^n$, we aim to extract clean audio signals $Y = \{y_i\}_{i=1}^n$ by learning a mapping $f$, and leverage $f(X) \approx Y$. In the Fourier frequency domain, we convert the audio denoising to an image generation task. Given the noisy audio complex spectrogram images (refer to images for the remaining paper) $\mathcal{I}_\mathcal{N} = \{I_{N_i}\}_{i=1}^n$ using STFT($X$) and clean audio complex images $\mathcal{I} = \{I_i\}_{i=1}^n$ using STFT($Y$), we also aim to find a function $F$ such that $F(\mathcal{I}_\mathcal{N}) \approx \mathcal{I}$, where $F(\mathcal{I}_\mathcal{N})$ are the generated complex images.

### 3.3 PRELIMINARY

Our model is inspired by the denoising diffusion probabilistic model (DDPM) (Ho et al., 2020), which first adds noise to the image through the forward diffusion process, then removes noise using the reverse process. The diffusion model consists of T steps. The forward process has the steps $t \in \{0, 1, ..., T\}$, and the reverse process has steps $t \in \{T, T-1, ..., 0\}$. The diffusion model iteratively adds noises to clean images and recovers clean images through the Markov chain. We first review the basic formulas of the diffusion model.

#### 3.3.1 FORWARD DIFFUSION PROCESS

In the forward process $q$, given a clean image $I^0 = I$, it can be formulated as follows:

$$q(I^{1:T}|I^0) = \prod_{t=1}^T q(I^t|I^{t-1}), \tag{2}$$

where $T$ is the number of steps and $I^1, \cdots I^T$ are latent variables. At each step of the forward process, Gaussian noise is added to the data according to the variance:

$$q(I^t|I^{t-1}) = \mathcal{N}(I^t; \sqrt{1-\beta_t}I^{t-1}, \beta_t\mathbf{I}), \quad q(I^t|I^0) = \mathcal{N}(I^t, \sqrt{\overline{\alpha}_t}I^0, (1-\overline{\alpha}_t)\mathbf{I}), \tag{3}$$

where $\mathbf{I}$ is the identity matrix, $\beta_t$ is a hyperparameter sequence that describes how much noise is added to each time step. $\alpha_t := 1 - \beta_t$ and $\overline{\alpha}_t := \prod_{s=1}^t \alpha_s$. With the reparametrization trick, we can write $I^t$ as a function of $I^0$:

$$I^t = \sqrt{\overline{\alpha}_t}I^0 + \sqrt{1-\overline{\alpha}_t}\epsilon, \quad \text{with } \epsilon \sim \mathcal{N}(0, \mathbf{I}). \tag{4}$$

#### 3.3.2 REVERSE PROCESS

The reverse process $p$ is parameterized by $\theta$ and defined by

$$p_\theta(I^{0:T-1}|I^T) = \prod_{t=1}^T p_\theta(I^{t-1}|I^t). \tag{5}$$

Starting from $p_\theta(I^T) = \mathcal{N}(I^T; 0, \mathbf{I})$, the reverse process converts the latent variable $p_\theta(I^T)$ to $p_\theta(I^0)$. At each step, it takes small Gaussian steps as follows:

$$p_\theta(I^{t-1}|I^t) = \mathcal{N}(I^{t-1}; \mu_\theta(I^t, t), \Sigma_\theta(I^t, t)). \tag{6}$$

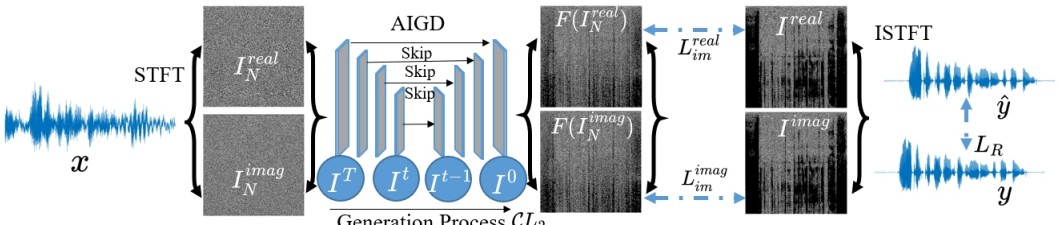

Figure 1: The schematic diagram of our AIGD model. We first apply STFT to convert audio signal $x$ into complex images (real image $I_N^{real}$ and imaginary image $I_N^{imag}$). Then, we feed them into our AIGD model ($F$, a complex U-Net architecture) and get generated real $F(I_N^{real})$ and imaginary $F(I_N^{imag})$ images by optimizing a complex L2 loss $\mathcal{C}L_2$. Finally, we minimize the image quality check loss $L_{im}^{total} = L_{im}^{imag} + L_{im}^{real}$, and audio reconstruction loss $L_R$.

We can predict $I^{t-1}$ from $I^t$ with

$$I^{t-1} = \frac{1}{\sqrt{\alpha_t}}(I^t - \frac{1 - \alpha_t}{\sqrt{1 - \overline{\alpha_t}}}\epsilon_\theta(I^t, t)) + \sigma_t z, \quad \text{with } z \sim \mathcal{N}(0, \mathbf{I}). \tag{7}$$

The evidence lower bound (ELBO) in DDPM is defined as:

$$D_{KL}(q(I^T|I^0)||p(I^T)) + \sum_{t=2}^{T} D_{KL}(q(I^{t-1}|I^t, I^0)||p_\theta(I^{t-1}|I^t)) - \log p_\theta(I^0|I^1). \tag{8}$$

### 3.4 AUDIO IMAGE GENERATION

Our method modifies the diffusion model by removing the unnecessary forward process and proposes an image generation process.

#### 3.4.1 AUDIOS AND IMAGES CONVERSION

Given an input batch of tensor $T = [N, C, H, W]$ using STFT($b(X)$), where $b$ is the batch size(*e.g.*, take $b = 32$ samples of $X$), $N$ is the number of samples in the batch; $C$ is the channel size ($C = 1$ if the audio is a single track, and $C = 2$ if the audio is dual tracks, we chose $C = 1$ to avoid some audios only have a single channel.); $H$ and $W$ are the height and width of the image (note that the tensor $T = A + jB \in \mathbb{C}^{N \times C \times H \times W}$ is a complex matrix, $A$ is the real part, and $B$ is the imaginary part), we define the basic deep learning operations $O$ as:

$$O(T) = O(A) + jO(B), \tag{9}$$

where $j$ is the square root of $-1$ and $O$ can be common deep learning layers. Therefore, we treat the real part and the imaginary part as separate images. Given any deep-learning image generation models $\mathcal{M}$, we could estimate the denoised images as:

$$\mathcal{M}(T) = \mathcal{M}(A) + j\mathcal{M}(B). \tag{10}$$

Therefore, we convert audio denoising into an image generation task. Our AIGD model is a complex U-Net architecture. The U-Net part is the same as the DDPM model, while our AIGD model supports the complex tensors as inputs and outputs using Eq. (10). AIGD model $F \in \mathcal{M}$, and we can retrieve the denoised audio $\hat{Y} = \text{ISTFT}(F(T))$, where ISTFT is the inverse STFT.

#### 3.4.2 IMAGE GENERATION PROCESS

Unlike the previous DDPM model, the noise audio images ($\mathcal{I_N} = \text{STFT}(X)$) already contain noises given $X$ are noisy audio signals. Hence, we remove the forward diffusion process and focus on the reverse process, naming it an image generation process. Given unknown clean audio image data distribution $p_{data}(\mathcal{I})$, the image generation process aims to remove the noise from noisy images $\mathcal{I_N}$ through generating high-quality denoised images $F(\mathcal{I_N})$, where $\mathcal{I}$ and $\mathcal{I_N}$ are both complex tensors. Given a clean image $I$ and a noisy image $I_N$, there exists a noisy model $\mathbb{N}$ that is denoted as follows:

$$I_N = \mathbb{N}(I; \epsilon_N, \theta_N), \tag{11}$$

where $I \in \mathcal{I}$, $I_N \in \mathcal{I}_\mathcal{N}$, $\epsilon_N$ is the noise and $\theta_N$ represents learned parameters of model $\mathbb{N}$. Typically, the supervised denoising model is trained by the following loss function:

$$L(\theta) = \mathbb{E}_{I,I_N}[e(I, F(I_N; \theta))], \tag{12}$$

where $e$ is the error and $F(\cdot; \theta)$ represents a model. Given noisy images are conditioned on clean images with unknown noisy model $\mathbb{N}$, we aim to estimate the posterior distribution of $p(I|I_N)$. Supposing our model $F$ is represented by $p_\theta(I^0|I^T)$, we define the generation process as a Markov chain as follows,

$$p_\theta(I^t|I^{t+1}, \cdots, I^T) = p_\theta(I^t|I^{t+1}, I^T), \tag{13}$$

where $I^0 = I$ is the clean image, $I^T = I_N$ is the noisy image, and $t = 0, \cdots, T-1$. We have the following property:

$$p_\theta(I|I_N) = p_\theta(I^0|I^T) = \int_{I^{1:T-1}} p_\theta(I^0, I^1, \cdots, I^{T-1}|I^T) dI^1 \cdots dI^{T-1} = \int_{I^{1:T-1}} \prod_{t=0}^{t=T-1}$$
$$p_\theta(I^t|I^{t+1}, \cdots, I^T) dI^1 \cdots dI^{T-1} = \int_{I_{1:T-1}} \prod_{t=0}^{t=T-1} p(I^t|I^{t+1}, I^T) dI^1 \cdots dI^{T-1}, \tag{14}$$

where $I^{1:T-1}$ is the abbreviation for $I^1, \cdots, I^{T-1}$. From Eq. (14), we can iteratively sample $p_\theta(I^t|I^{t+1}, I^T)$ to estimate the posterior distribution $p(I|I_N)$. In Eq. (11), **we assume that $\mathbb{N}$ is an arbitrary noise function, that we could use finite Gaussian mixture models to approximate it.** We have

$$I^T = I^0 + \sum_{t=0}^{T-1}(\sigma_{t+1}^2 - \sigma_t^2)z_t, \quad z_t \sim \mathcal{N}(\mu_{rt}, \sigma_{rt}), \tag{15}$$

where $\mathcal{N}(\mu_{rt}, \sigma_{rt})$ is a standard Gaussian distribution, with random generated $\mu_r$ and $\sigma_r$ in each step $t$ and $(\sigma_{t+1}^2 - \sigma_t^2)$ is the coefficient of each $z_t$. Although there is no forward diffusion process, we assume that the noise is continuously added through a Markov chain. Let $0 = \sigma_0 < \sigma_1 < \cdots \sigma_{T-1} < \sigma_T = \sigma$, then $\{\sigma_t\}$ is monotonically increased. We can get each image $I^{t+1}$ by

$$I^{t+1} = I^t + (\sigma_{t+1}^2 - \sigma_t^2)z_t, \quad z_t \sim \mathcal{N}(\mu_{rt}, \sigma_{rt}), \tag{16}$$

where $t = 0, 1, \cdots, T-1$, and we have

$$I^t = I^0 + \sum_{t=0}^{T-1}(\sigma_{t+1}^2 - \sigma_t^2)z_t, \quad z_t \sim \mathcal{N}(\mu_{rt}, \sigma_{rt}), \tag{17}$$

where $t = 0, 1, \cdots, T$. From Eq. (16), we can get

$$p(I^t|I^{0:t-1}) = p(I^t|I^{t-1}). \tag{18}$$

Therefore, $I^t$, where $t = 0, 1, \cdots, T$ are a Markov chain. Eventually, when $t = T$, the Gaussian noise is continuously added as in Eq. (15). We can further derive

$$p(I^{1:T-1}|I^0, I^T) = \prod_{t=1}^{T-1} p(I^t|I^0, I^{t+1}), \quad p(I^t|I^{t+1:T} = p(I^t|I^{t+1})). \tag{19}$$

Similar to Sec. 3.3, we derive the evidence lower bound objective (ELBO) as follows:

$$L = \mathbb{E}_{I^0,I^T}\left\{ -\log p_\theta(I^0|I^N) \right\} \leq \mathbb{E}_{p^{0:T}}\left\{ -\log \frac{\prod_{t=0}^{t=T-1} p_\theta(I^t|I^{t+1}, I^T)}{p(I^{1:T-1}|I^0, I^T)} \right\}$$
$$= \sum_{t=1}^{T-1} \mathbb{E}_{p^{0,t+1,T}}[D_{KL}(p(I^t|I^0, I^{t+1})||p_\theta(I^t|I^{t+1}, I^T))] + \mathbb{E}_{p^{0,1,T}}[-\log p_\theta(I^0|I^1, I^T)]. \tag{20}$$

From Eq. (19), given $I^{t+1}$, $I^t$ is not dependent on $I_T$ when $t < T-1$, we further assume $p_\theta(I^t|I^{t+1}, I^T) = p_\theta(I^t|I^{t+1})$. Eq. (20) can be denoted as:

$$L \leq \sum_{t=1}^{T-1} \mathbb{E}_{p^{0,t+1,T}}[D_{KL}(p(I^t|I^0, I^{t+1})||p_\theta(I^t|I^{t+1}))] + \mathbb{E}_{p^{0,1,T}}[-\log p_\theta(I^0|I^1)]. \tag{21}$$

For the first term, we consider the analysis of $p(I^t|I^0, I^{t+1})$, we have

$$p(I^t|I^0, I^{t+1}) \sim \mathcal{N}(\tilde{\mu}_t, \tilde{\sigma}_t \mathbf{I}), \quad t = 0, \cdots, T-1, \tag{22}$$

where
$$\tilde{\mu}_t = \frac{\sigma_t^2}{\sigma_{t+1}^2} I^{t+1} + \frac{\sigma_{t+1}^2 - \sigma_t^2}{\sigma_{t+1}^2} I^0, \quad \tilde{\sigma}_t = \frac{\sigma_t}{\sigma_{t+1}} \sqrt{\sigma_{t+1}^2 - \sigma_t^2}. \tag{23}$$

We can also assume that:

$$p_\theta(I^t|I^{t+1}) \sim \mathcal{N}(\mu_{\theta,t+1}(I^{t+1}), \tilde{\sigma}_t \mathbf{I}), \quad \text{where}$$

$$\mu_{\theta,t+1}(I^{t+1}) = \frac{\sigma_t^2}{\sigma_{t+1}^2} I^{t+1} + \frac{\sigma_{t+1}^2 - \sigma_t^2}{\sigma_{t+1}^2} F(I^{t+1}, t+1; \theta). \tag{24}$$

$F$ is our AIGD model and $F(I^{t+1}, t+1; \theta)$ is the predicted denoised image given the input of $(I^{t+1}, t+1)$. The proofs can be found in the supplementary material. Given $p(I^t|I^0, I^{t+1})$ and $p_\theta(I^t|I^{t+1})$ have the same covariance matrix, for the first term in Eq. (21), we have

$$D_{KL}(p(I^t|I^0, I^{t+1})||p_\theta(I^t|I^{t+1})) = ||\mu_{\theta,t+1}(I^{t+1}) - \tilde{\mu}_t||_2^2, \tag{25}$$

where $|| \cdot ||_2^2$ is L2 norm. Combining Eq. (25) with Eq. (23), Eq. (24), and neglecting the constant term, minimizing Eq. (25) is equivalent to minimizing the follows,

$$||F(I^{t+1}, t+1; \theta) - I^0||_2^2. \tag{26}$$

Next, for the second term $\mathbb{E}_{p^{0,1,T}}[-\log p_\theta(I^0|I^1)]$ of Eq. (21), we train $p_\theta$ using $\mathbb{E}_{p^{0,1,T}}[F(I^1, 1; \theta) - I^0]$. Considering the above analysis, our main loss function in the generation process is:

$$\mathcal{C}L_2 = \sum_{t=0}^{T-1} ||F(I^{t+1}, t+1; \theta) - I^0||_2^2$$

$$= \sum_{t=0}^{T-1} (||F(I^{t+1}, t+1; \theta).real - I^0.real||_2^2 + ||F(I^{t+1}, t+1; \theta).imag - I^0.imag||_2^2), \tag{27}$$

where $\mathcal{C}L_2$ is the complex L2 norm, $I^0 = I$ is the clean image. $real$ and $imag$ are the real and imaginary parts of complex tensors, respectively. Through the generation process, given noisy input image $I_N$, our AIGD model can predict the denoised image $F(I_N)$ that is similar to clean image $I$.

### 3.5 IMAGE QUALITY CHECK

Eq. (27) only enforces the predicted images to be close to clean images, while the quality of generated images is not explored. To improve the generated images' quality, we developed an image quality check module. We impose a complex absolute structural similarity loss $L_S$ as follows:

$$L_S = 1 - abs(\text{SSIM}(F(I_N), I))$$
$$= \{1 - abs(\text{SSIM}(F(I_N).real, I.real)) + 1 - abs(\text{SSIM}(F(I_N).imag, I.imag)\}/2, \tag{28}$$

where $SSIM$ is the structural similarity index measure (Wang et al., 2004). The range of the $L_S$ is from 0 to 1, where 0 indicates high similarity between images and 1 means they are not similar. As shown in Fig. 1, we could get two different images given one audio: real image and imaginary image. Eventually, the image quality check module loss consists of the above image minimizations and is defined as:

$$L_{im}^{total} = L_{im}^{real} + L_{im}^{imag}, \tag{29}$$

where $L_{im}^{real} = \sum_{t=0}^{T-1} ||F(I^{t+1}, t+1; \theta).real - I^0.real||_2^2 + 1 - abs(\text{SSIM}(F(I_N).real, I.real))$, and $L_{im}^{imag} = \sum_{t=0}^{T-1} ||F(I^{t+1}, t+1; \theta).imag - I^0.imag||_2^2 + 1 - abs(\text{SSIM}(F(I_N).imag, I.imag))$. Hence, our AIGD model can improve the quality of generated denoised images.

### 3.6 AUDIO RECONSTRUCTION

After obtaining the output from the decoder layers from the AIGD model, we could apply ISTFT to get the reconstructed audio as $\hat{y}$. We propose an SDR loss to evaluate the quality of $\hat{y}$. The SDR is defined as: $SDR(\hat{y}, y) = 10 \log_{10} \frac{||y||^2}{||\hat{y}-y||^2}$. We defined the SDR loss as:

$$L_{SDR} = const_{upper} - SDR(\hat{y}, y), \tag{30}$$

where $const_{upper}$ is the upper bound constant value, we set it as 30. A detailed ablation study is shown in Sec. 4.2. Therefore, we could ensure that the SDR loss keeps decreasing during the training.

### 3.7 OBJECTIVE FUNCTION

The architecture of our proposed AIGD model is shown in Fig. 1. Considering all loss functions in Sec. 3.5 and Sec. 3.6, our model minimizes the following objective function

$$L = \alpha L_{im}^{total} + (1 - \alpha)L_R, \tag{31}$$

where $\alpha$ is the balance factor between all image loss and audio reconstruction loss. This objective function enables us first to get high-quality generated images, then acquire a better reconstructed denoised audio. Our training procedures are described in Alg.1.

---

**Algorithm 1** The training process of AIGD model

---

1: **Input:** Noise audio signal $x$, clean audio signal $y$, and AIGD model $F(\cdot, \cdot, \theta)$.
2: **Output:** Trained image generator $F(\cdot, \cdot, \theta)$
3: Generate noisy audio image $I_N$ and clean audio images $I$ using STFT($x$) and STFT($y$)
4: **while** $\theta$ is not converged **do**
5:     **for** $t = T - 1, \cdots, 1$ **do**
6:         Sample $I^t$ using Eq. (24)
7:     **end for**
8:     Predict denoised image $\hat{I} = F(I^1, 1, \theta)$
9:     Compute gradient using the objective function in Eq. (31)
10:    Update $\theta$ by gradient
11: **end while**

---

## 4 EXPERIMENTS

**Dataset:** we first evaluate our AIGD model on the **VoiceBank-DEMAND** (Valentini-Botinhao et al., 2017) dataset. It is a synthetic dataset created by mixing up clean speech and noise. The training set contains 11,572 utterances (9.4h), and the test set contains 824 utterances (0.6h). The lengths of utterances range from 1.1s to 15.1s, with an average of 2.9s. We report six metrics of our model, including perceptual evaluation of speech quality (PESQ), short-time objective intelligibility (STOI), prediction of the signal distortion (CSIG), prediction of the background intrusiveness (CBAK), prediction of the overall speech quality (COVL), and structural similarity index measure (SSIM). Evaluation of the **DNS challenge (INTERSPEECH 2020)** and **BirdSoundsDenoising** (Zhang & Li, 2023) datasets and **computation time** can be found in the supplementary material.

### 4.1 IMPLEMENTATION DETAILS

AIGD model is a complex U-Net architecture that enables complex tensors as inputs and outputs (U-Net is from DDPM Ho et al. (2020)). During the training, we set batch size = 2, training iteration = 1,000, learning rate = 0.001, time step T = 1,000, and $\alpha = 0.5$ with an Adam optimizer on a 48G RTX A6000 GPU using PyTorch. To represent more details of audio, we consider image size as $[512 \times 512]$. In PyTorch's STFT function, we utilized 1000-point Hamming as the window function, the size of Fourier transform $n\_fft = 1023$, which will output fixed image height as 512. Since the length of each audio can be different, we set the distance between neighboring sliding window frames as $hop\_length = int(length(y_t)/512)$, where $length(y_t)$ is the length of each audio, which will approach the width size of 512. We then resize the input image dimensions for the AIGD model as gray-scale image $[1 \times 512 \times 512]$. During the ISTFT, the same parameters will be used to reconstruct denoised audio signals. To show the effectiveness of our proposed AIGD model in image generation, we also report the results of three frequently used image generation methods (CycleGAN (Zhu et al., 2017), Pix2Pix (Isola et al., 2017) and DDPM (Ho et al., 2020)) and one image denoising method (DnCNN (Zhang et al., 2017)). For these four methods, we use the same training algorithm in Alg. 1 except that the U-Net in our AIGD model is replaced with these four different models[1].

### 4.2 RESULTS

Tab. 1 shows the comparison results of the VoiceBank-DEMAND dataset. For nine baseline models, we also get the generated images following the same STFT parameters in Sec. 4.1. We reported the

---

[1]Source code will be made available upon acceptance.

Table 1: Comparison results on the VoiceBank-DEMAND dataset. "N/A": not applicable.

| Methods | Domain | PESQ | STOI | CSIG | CBAK | COVL | SSIM |
|---|---|---|---|---|---|---|---|
| PGGAN (Li et al., 2022) | T | 2.81 | 0.944 | 3.99 | 3.59 | 3.36 | 0.56 |
| DCCRGAN (Huang et al., 2022) | TF | 2.82 | 0.949 | 4.01 | 3.48 | 3.40 | 0.65 |
| S-DCCRN (Lv et al., 2022) | TF | 2.84 | 0.940 | 4.03 | 2.97 | 3.43 | 0.62 |
| DCU-Net (Choi et al., 2019) | TF | 2.93 | 0.930 | 4.10 | 3.77 | 3.52 | 0.67 |
| MetricGAN+ (Fu et al., 2021) | TF | 3.15 | 0.927 | 4.14 | 3.12 | 3.52 | 0.78 |
| TSTNN (Wang et al., 2021) | T | 2.96 | 0.950 | 4.33 | 3.53 | 3.67 | 0.76 |
| MANNER (Park et al., 2022) | T | 3.21 | 0.950 | 4.53 | 3.65 | 3.91 | 0.80 |
| DnCNN (Zhang et al., 2017) | TF | 2.76 | 0.943 | 3.96 | 3.41 | 3.30 | 0.59 |
| CycleGAN (Zhu et al., 2017) | TF | 2.87 | 0.948 | 4.08 | 3.70 | 3.48 | 0.65 |
| Pix2Pix (Isola et al., 2017) | TF | 2.96 | 0.950 | 4.34 | 3.55 | 3.69 | 0.79 |
| DDPM (Ho et al., 2020) | TF | 3.03 | 0.951 | 4.17 | 3.51 | 3.72 | 0.79 |
| **AIGD** | TF | **3.52** | **0.971** | **4.82** | **3.99** | **4.25** | **0.87** |

extra structure similarity (SSIM) between the generated image and the ground truth (mean SSIM of real and imaginary images). Our proposed AIGD model achieves the highest performance in all six metrics. Especially the SSIM metric is much higher than all other methods. Note that the key to success here is that our AIGD model directly denoises the noisy images and removes the unnecessary forward diffusion process of the original DDPM model, which is the reason that AIGD is better than the re-implemented DDPM model. To check the quality of generated images, we compare generated complex images with the five models using two samples in Fig. 2. We also list their mean SSIM and PESQ scores. The generated real and imaginary images of the AIGD model are close to ground truth, while the other five models contain many noise areas. Although the images of DDPM are similar to those of our AIGD model, they contain more black areas. As for Pix2Pix and CycleGAN models, they both contain many noisy areas that are not removed, which means that these two models cannot generate clean images. DnCNN removes too many areas in the generated images. It is a surprise that it has a better performance than the DDPM model in the first example, given the ground truth contains more black areas. For MANNER, it is not an image generation method. Once we use STFT to convert it to the frequency domain, it shows a different pattern of images. From these experiments, we can conclude that a higher SSIM score (better-generated images) will output better reconstructed audio signals and eventually achieve better audio-denoising performance.

**Parameters analysis.** In Eq. (31), $\alpha$ balances the image quality check loss and audio reconstruction loss. We first conduct the parameter analysis of $\alpha$, which is selected from $\{0, 0.1, 0.2, 0.3, 0.4, 0.5, 0.6, 0.7, 0.8, 0.9, 1\}$. "0" means that we only minimize the SDR loss, while "1" means that we only minimize the image check model. As shown in Fig. 3a, we chose $\alpha = 0.5$ as our best hyperparameter since PESQ achieves the highest value. Next, we explore the relationship among PESQ, SDR, and SSIM metrics. We defined complex absolute structural similarity loss $L_S$ and SDR loss $L_R$ in Eqs. (28) and (30), the lower of these two loss functions, the better the denoised results. While the higher SDR, SSIM, and PESQ, the better the results since they measure the closeness between the predictions and ground truth. As shown in Fig. 3b, with the increasing number of iterations, SSIM and PESQ converged fast and approached the highest value (1 and 4.5). Although there are some oscillations at the end of the SDR value, it is still converged, and the highest value is 25.39. Hence, we set the upper bound of SDR loss in Eq. (30) as 30.

Table 2: Ablation study of different loss functions

| Methods | 2 | S | R | 2+S | S+R | 2 + R | 2+S+R |
|---|---|---|---|---|---|---|---|
| PESQ | 3.25 | 2.50 | 2.89 | 3.28 | 2.95 | 3.31 | **3.52** |

**Ablation study.** To demonstrate the effectiveness of the proposed three loss functions: complex L2 norm (2), complex absolute structure similarity (S), and SDR (R), we conduct an ablation study with respect to each loss in Tab. 2. "+" means combining loss functions together. We observe that with the increasing number of loss functions, the robustness of our model keeps improving. The usefulness of loss functions is ranked as 2 > R > S. Therefore, the proposed audio image generation denoising approach is effective in improving performance, and different loss functions are helpful and important in minimizing the error between predictions and ground truth.

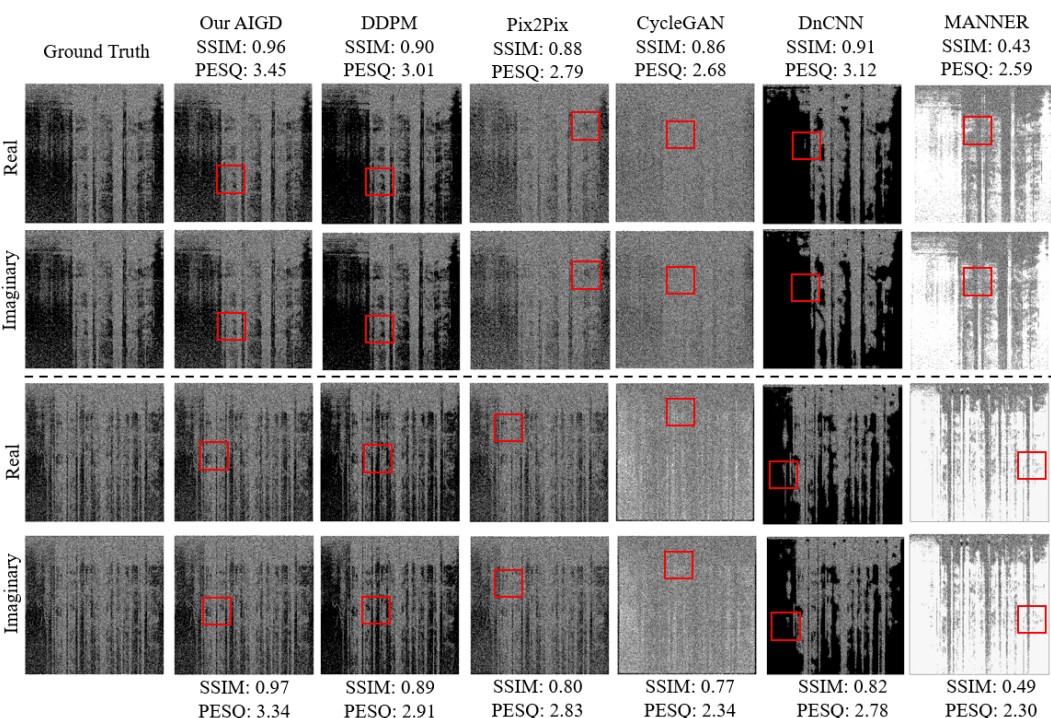

Figure 2: Comparison of generated images of our AIGD and five methods of real and imaginary images in two samples (separated by a dashed line). Red squares show the different areas between real and imaginary images. More results are shown in the supplementary material.

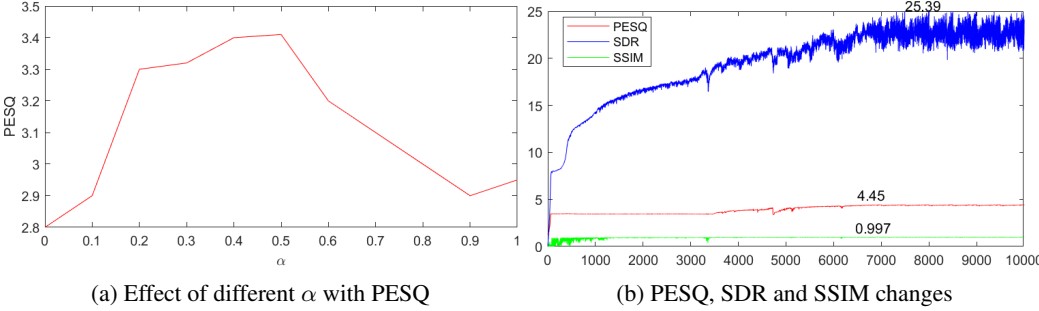

(a) Effect of different $\alpha$ with PESQ

(b) PESQ, SDR and SSIM changes

Figure 3: (a): parameter analysis for $\alpha$ with PESQ. (b): PESQ, SDR, and SSIM change with the increase of training iterations (10,000). The highest values are plotted for each line.

**Reflection.** From Tab. 1 and Fig. 2, we can conclude that our proposed AIGD model achieves state-of-the-art performance, which also demonstrates the superiority of the proposed architecture and novel loss functions. However, we can still observe some missing areas of generated real and imaginary images in Fig. 2, which indicates that there is still space to further improve our model. A more effective image check loss function can be developed in future work. One weakness of our model is that it requires high GPU memory to train the model. Our AIGD model has 35M parameters, and it took around 2.4 minutes to train per audio but less than 1.2 seconds (per audio) for inference.

## 5 CONCLUSION

In this paper, we convert the audio denoising task into an image generation problem. We first develop an image generation progress to estimate the posterior distribution of clean complex images conditioned on noisy complex images. We then impose complex L2 and complex absolute structure similarity losses to generate high-quality images and develop an SDR loss to minimize the difference between denoised audio and clean audio. Extensive experiments demonstrate our proposed AIGD model outperforms state-of-the-art models.

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

## A  APPENDIX

## B  PROOFS

### B.1  PROOF OF THE FIRST PART OF EQ. (19) OF THE MAIN PAPER IN SEC. 3.4.2

$$p(I^{1:T-1}|I^0, I^T) = \prod_{t=1}^{T-1} p(I^I|I^0, I^{t+1})$$

*Proof.* According to Eq. (18) of the main paper, we have the following deviation:

$$p(I^{1:T-1}|I^0, I^T)$$

$$= \prod_{t=1}^{T-1} p(I^t|I^0, I^{t+1:T})$$

$$= \prod_{t=1}^{T-1} \frac{p(I^t, I^{t+2:T}|I^0, I^{t+1})}{p(I^{t+2:T}|I^0, I^{t+1})}$$

$$= \prod_{t=1}^{T-1} \frac{p(I^{t+2:T}|I^0, I^{t+1}, I^t)p(I^t|I^0, I^{t+1})}{p(I^{t+2:T}|I^0, I^{t+1})}$$

$$= \prod_{t=1}^{T-1} \frac{p(I^{t+2:T}|I^{t+1})q(I^t|I^0, I^{t+1})}{p(I^{t+2:T}|I^{t+1})}$$

$$= \prod_{t=1}^{T-1} p(I^t|I^0, I^{t+1})$$

Eq. (18) is applied in the fourth equation.

## B.2 PROOF OF THE SECOND PART OF EQ. (19) OF THE MAIN PAPER IN SEC. 3.4.2

$$p(I^t|I^{t+1:T}) = p(I^t|I^{t+1})$$

*Proof.*

$$p(I^t|I^{t+1:T})$$

$$= \frac{p(I^t, I^{t+2:T}|I^{t+1})}{p(I^{t+2:T}|I^{t+1})}$$

$$= \frac{p(I^t|I^{t+1})p(I^{t+2:T}|I^{t+1}, I^t)}{p(I^{t+2:T}|I^{t+1})}$$

$$= \frac{p(I^t|I^{t+1})p(I^{t+2:T}|I^{t+1})}{p(I^{t+2:T}|I^{t+1})}$$

$$= p(I^t|I^{t+1})$$

Eq. (18) is applied in the third equation.

## B.3 THE PROOF OF EQ. (20) OF THE MAIN PAPER IN SEC. 3.4.2

$$L \leq \mathbb{E}_{p^{0:T}}\left\{ -\log\frac{\prod_{t=0}^{t=T-1} p_\theta(I^t|I^{t+1}, I^T)}{p(I^{1:T-1}|I^0, I^T)} \right\}$$

*Proof*

$$L = \mathbb{E}_{I^0, I^T} \{ -\log p_\theta(I^0|I^T) \}$$

$$= \mathbb{E}_{I^0, I^T} \left\{ -\log \int_{I^1, \cdots, I^{T-1}} p_\theta(I^0, I^1, \cdots, I^{T-1}|I^T) dI^1, \cdots, dI^{T-1} \right\}$$

$$= \mathbb{E}_{I^0, I^T} \left\{ -\log \int_{I^1, \cdots, I^{T-1}} \frac{p_\theta(I^0, I^1, \cdots, I^{T-1}|I^T)}{p(I^1, \cdots, I^{T-1}|I^0, I^T)} p(I^1, \cdots, I^{T-1}|I^0, I^T) dI^1, \cdots, dI^{T-1} \right\}$$

$$\leq \mathbb{E}_{I^0, I^T} \left\{ -\int_{I^1, \cdots, I^{T-1}} p(I^1, \cdots, I^{T-1}|I^0, I^T) \log \frac{p_\theta(I^0, I^1, \cdots, I^{T-1}|I^T)}{p(I^1, \cdots, I^{T-1}|I^0, I^T)} dI^1, \cdots, dI^{T-1} \right\}$$

$$= \mathbb{E}_{I^0, I^T} \left\{ \mathbb{E}_{I^1, \cdots, I^{T-1}|I^0, I^T} \left\{ -\log \frac{p_\theta(I^0, I^1, \cdots, I^{T-1}|I^T)}{p(I^1, \cdots, I^{T-1}|I^0, I^T)} \right\} \right\}$$

$$= \mathbb{E}_{I^0, I^1, \cdots, I^{T-1}, I^T} \left\{ -\log \frac{p_\theta(I^0, I^1, \cdots, I^{T-1}|I^T)}{p(I^1, \cdots, I^{T-1}|I^0, I^T)} \right\}$$

$$= \mathbb{E}_{p^{0:T}} \left\{ -\log \frac{p_\theta(I^{0:T-1}|x^T)}{p(I^{1:T-1}|I^0, I^T)} \right\}$$

$$= \mathbb{E}_{p^{0:T}} \left\{ -\log \frac{\prod_{t=0}^{t=T-1} p_\theta(I^t|I^{t+1}, I^T)}{p(I^{1:T-1}|I^0, I^T)} \right\}$$

$$= \mathbb{E}_{p^{0:T}} \left\{ -\log \frac{\prod_{t=0}^{t=T-1} p_\theta(I^t|I^{t+1}, I^T)}{\prod_{t=1}^{T-1} p(I^{1:T-1}|I^0, I^T)} \right\}$$

$$= \mathbb{E}_{p^{0:T}} \left\{ -\sum_{t=1}^{T-1} \log \frac{p_\theta(I^t|I^{t+1}, I^T)}{p(I^t|I^0, I^T)} - \log p_\theta(I^0|I^1, I^T) \right\}$$

$$= \sum_{t=1}^{T-1} \mathbb{E}_{p^{0,t+1,T}} [D_{KL}(p(I^t|I^0, I^{t+1})||p_\theta(I^t|I^{t+1}, I^T))] + \mathbb{E}_{p^{0,1,T}} [-\log p_\theta(I^0|I^1, I^T)]$$

### B.4 PROOF OF EQ. (22) AND EQ. (23) OF THE MAIN PAPER IN SEC. 3.4.2

$$p(I^t|I^0, I^{t+1}) \sim \mathcal{N}(\tilde{\mu}_t, \tilde{\sigma}_t I), t = 0, 1, \cdots, T-1,$$

where,

$$\tilde{\mu}_t = \frac{\sigma_t^2}{\sigma_{t+1}^2} I^{t+1} + \frac{\sigma_{t+1}^2 - \sigma_t^2}{\sigma_{t+1}^2} I^0,$$

$$\tilde{\sigma}_t = \frac{\sigma_t}{\sigma_{t+1}} \sqrt{\sigma_{t+1}^2 - \sigma_t^2}$$

*Proof.* Firstly, we have the following equation according to the Bayesian rule.

$$p(I^t|I^0, I^{t+1}) = \frac{p(I^t, I^{t+1}|I^0)}{p(I^{t+1}|I^0)} = \frac{p(I^t|I^0)p(I^{t+1}|I^t)}{p(I^{t+1}|I^0)}.$$

Since $p(I^t|I^0)$, $p(I^{t+1}|I^t)$ and $p(I^{t+1}|I^0)$ all follow Gaussian distribution, we can derive the analytical form for $p(I^t|I^0, I^{t+1})$, which also follows Gaussian distribution and the parameters are derived as in Eq. (23).

## C GENERATED IMAGES VISUALIZATION AND COMPARISON

To show the effectiveness of our proposed AIGD model, we also list more generated complex images and compare them with three image generation methods: original DDPM (Ho et al., 2020), Pix2Pix (Isola et al., 2017), and CycleGAN (Zhu et al., 2017). We did not show the images of DnCNN (Zhang et al., 2017) and MANNER (Park et al., 2022) since their generated images have low quality.

As shown in Figures 4 to 6, our proposed AIGD model has the highest SSIM and PESQ scores. We can also find that the zoom-in red patch areas of our model are more similar to ground truth images than the other three methods in both real and imaginary images. The denoised audio signals seem slightly different from each other, and this is caused by the small changes in the audio signals that are difficult to visualize for human eyes. However, compared with the input noisy audio signal of the last row of each figure, we can find that denoised audios are much better than the noisy input audios. Note that all baseline methods: DDPM, Pix2Pix, and CycleGAN utilized the same architecture as our AIGD model, except that we replaced AIGD with the corresponding model. This indicates our proposed flowchart and loss functions are effective in improving the performance of audio denoising.

In Figure 4, the zoom-in patch of DDPM is close to our AIGD model, while it has many missing areas in both real and imaginary images. The Pix2Pix introduced some noises in the generated images, while CycleGAN contained many noises. Therefore, the ranking of generated images is: AIGD > DDPM > Pix2Pix > CycleGAN.

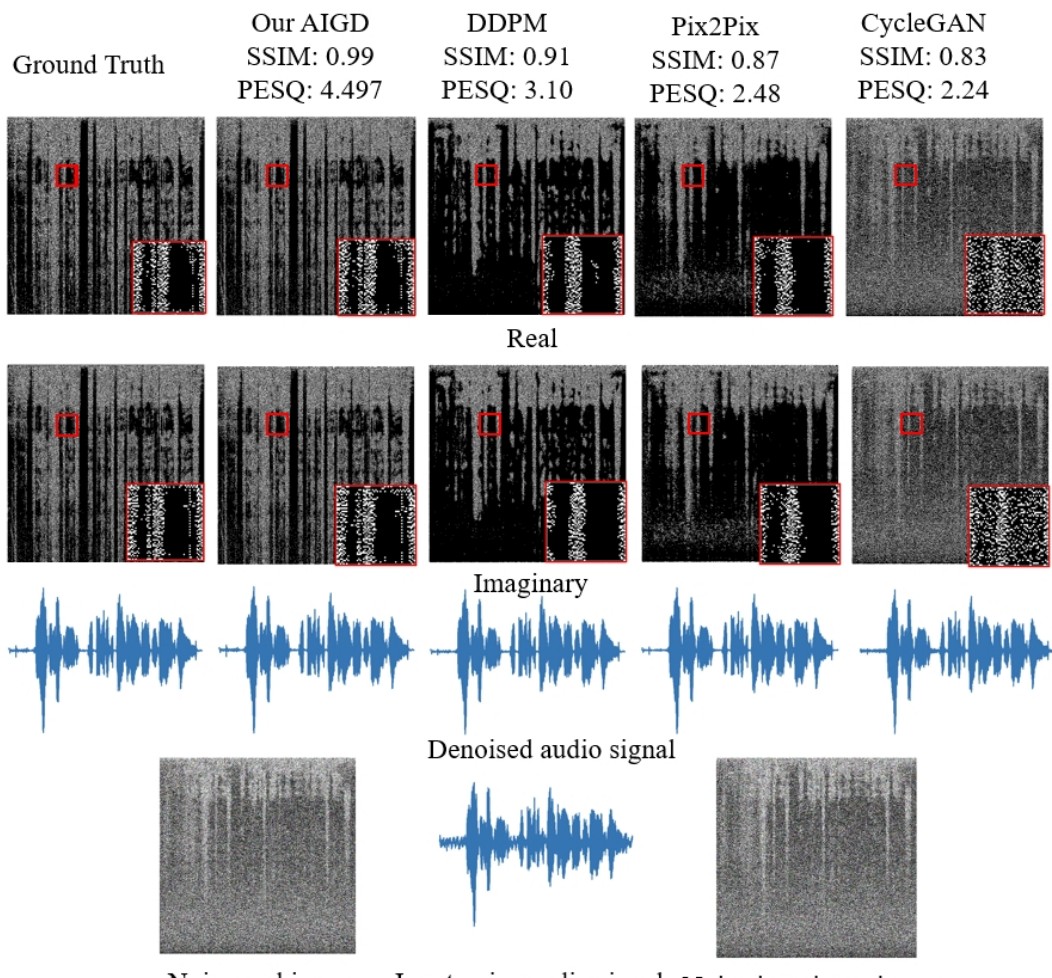

Figure 4: Results comparison of example one. The first row is the generated real image. The second is the generated imaginary image. The third row is the denoised audio signals of four methods. The last row shows the input noisy real and imaginary images and the corresponding noisy audio signal. The ground truth in the first column is the clean signal corresponding to the nosy audio signal in the last row. The small red path in each image corresponds to the zoom-in view in each big red right bottom area.

In Figure 5, although the zoom-in patch of our AIGD model is slightly different from the ground truth, it is much better than the other three methods. DDPM, Pix2Pix, and CycleGAN all contain many noises in the generated images.

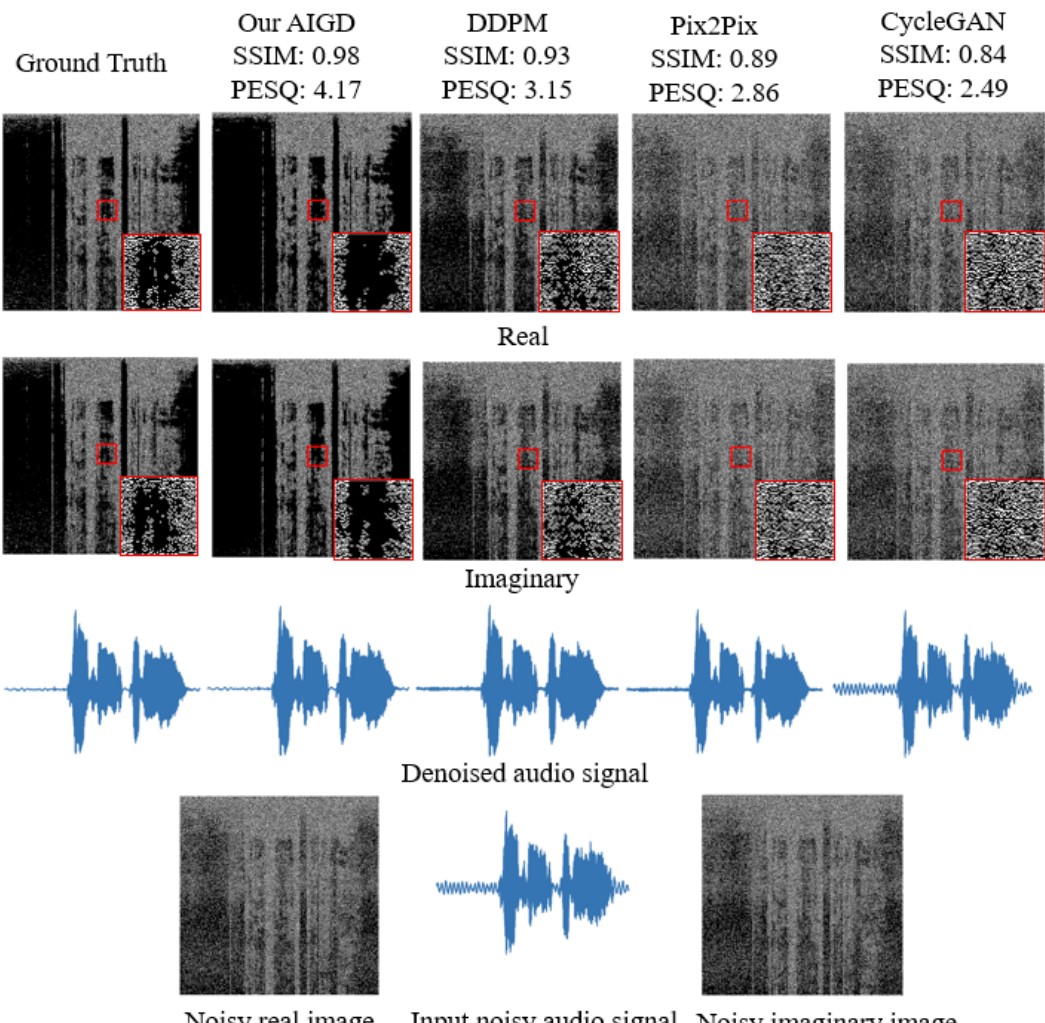

Figure 5: Results comparison of example two.

In Figure 6, our AIGD model also achieves the highest PESQ value even if the input audio signal is very noisy (the real and imaginary images contain many random noises). This further demonstrates that our AIGD is suitable for audio denoising, especially for noisy audio.

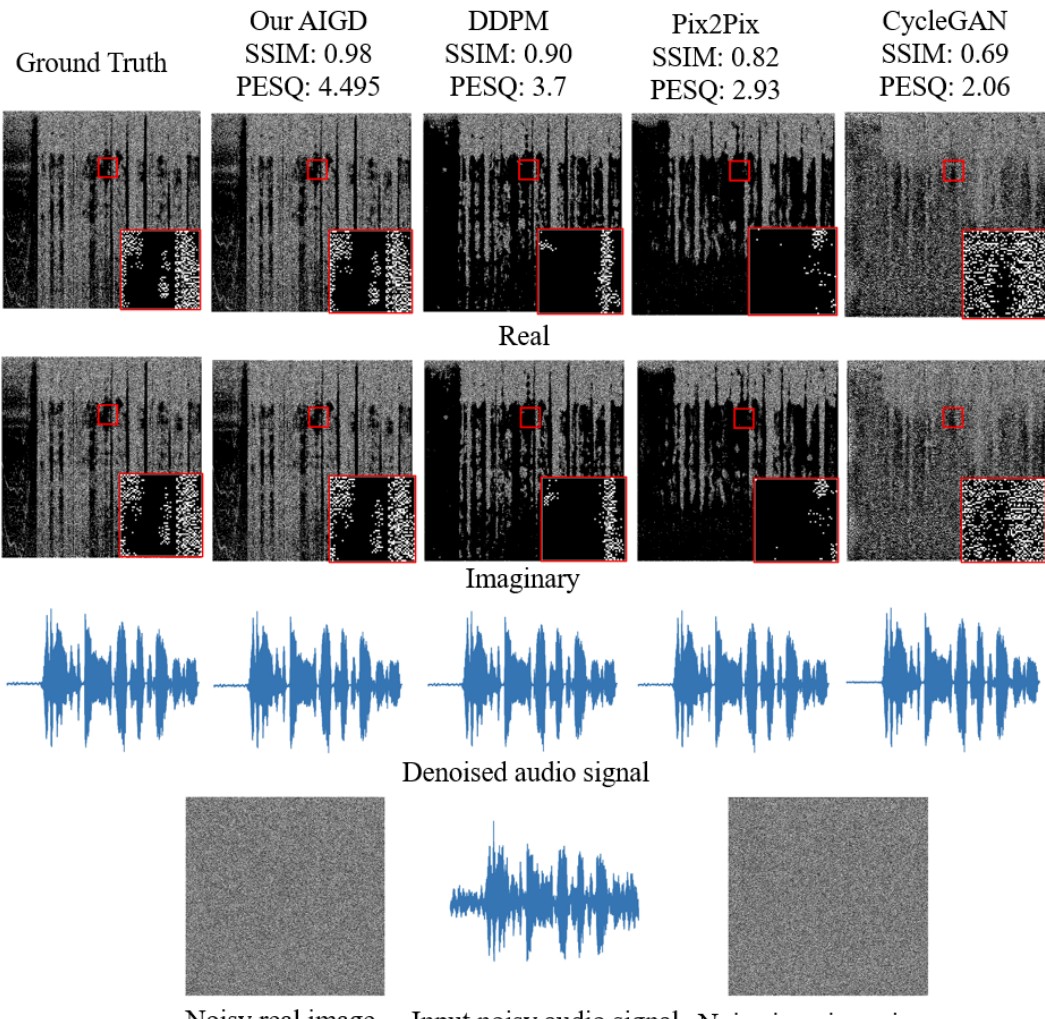

Figure 6: Results comparison of example three.

# D    EVALUATION ON DNS 2020 CHALLENGE DATASET

We also evaluate our AIGD model on the DNS challenge (INTERSPEECH 2020) dataset (Reddy et al., 2020). The clean speech set has 500 hours of clips from 2150 speakers. The noise dataset has 180 hours of clips from 150 classes. We follow the settings of (Hao et al., 2021), speech-noise mixture with dynamic mixing are simulated during the training. The test dataset of DNS Challenge includes two categories of synthetic clips, i.e., without and with reverberations. Each category has 150 noisy clips with SNR levels distributed between 0 dB and 20 dB. We use this test dataset for evaluation. We report four metrics, including wide-band perceptual evaluation of speech quality (WB-PESQ), narrow-band perceptual evaluation of speech quality (NB-PESQ), short-time objective intelligibility (STOI) and scale invariant signal-to-distortion ratio (SI-SDR).

As shown in Tab. 3, our AIGD model outperforms all other state-of-the-art models in all four metrics. Especially, the PESQ metrics are much higher than other models, which further reveals that our AIGD model achieves state-of-the-art performance in audio-denoising tasks.

Table 3: Comparison results on DNS 2020 challenge test dataset. "−" means not applicable.

| Methods | With Reverb | | | | Without Reverb | | | |
|---|---|---|---|---|---|---|---|---|
| | WB-PESQ | NB-PESQ | STOI | SI-SDR | WB-PESQ | NB-PESQ | STOI | SI-SDR |
| Noisy | 1.822 | 2.753 | 86.62 | 9.033 | 1.582 | 2.454 | 91.52 | 9.071 |
| NSNet (Xia et al., 2020) | 2.365 | 3.076 | 90.43 | 14.721 | 2.145 | 2.873 | 94.47 | 15.613 |
| DTLN (Westhausen & Meyer, 2020) | - | 2.700 | 84.68 | 10.530 | - | 3.040 | 94.76 | 16.340 |
| Conv-TasNet (Koyama et al., 2020) | 2.750 | - | - | - | 2.730 | - | - | - |
| DCCRN-E (Hu et al., 2020) | - | 3.077 | - | - | - | 3.266 | - | - |
| PoCoNet (Isik et al., 2020) | 2.832 | - | - | - | 2.748 | - | - | - |
| Sub-band Model (Li & Horaud, 2020) | 2.650 | 3.274 | 90.53 | 14.673 | 2.369 | 3.052 | 94.24 | 16.153 |
| FullSubNet (Hao et al., 2021) | 2.969 | 3.473 | 92.62 | 15.750 | 2.777 | 3.305 | 96.11 | 17.290 |
| FullSubNet+ (Chen et al., 2022b) | 3.218 | 3.666 | 93.84 | 16.810 | 2.982 | 3.504 | 96.69 | 18.340 |
| FS-CANet (Chen et al., 2022a) | 3.218 | 3.665 | 93.93 | 16.820 | 3.017 | 3.513 | 96.74 | 18.080 |
| **AIGD** | **3.381** | **3.912** | **95.02** | **18.213** | **3.351** | **4.013** | **98.31** | **20.123** |

# E    EVALUATION ON BIRDSOUNDSDENOISING DATASET

To test the performance on the real-world noisy dataset, we evaluate our proposed model on the BirdSoundsDenoising dataset (Zhang & Li, 2023). This dataset replaces the usual artificially added noise with natural noises, including wind, waterfalls, rain, etc. In particular, the dataset contains 14,120 audios from one second to fifteen seconds and is a large-scale dataset of bird sounds collected, containing 10,000/1,400/2,720 in training, validation, and testing datasets, respectively.

As shown in Tab. 4, we ignored the $F1$, $IoU$, and $Dice$ scores, given the original paper treated it as an image segmentation problem. Our AIGD model also achieves the highest SDR score than other models. Therefore, the AIGD model also has a better performance in a real-world bird sound denoising dataset.

Table 4: Results comparisons of different methods ($F1$, $IoU$, and $Dice$ scores are multiplied by 100. "$-$" means not applicable.

| Networks | Validation | | | | Test | | | |
|---|---|---|---|---|---|---|---|---|
| | $F1$ | $IoU$ | $Dice$ | $SDR$ | $F1$ | $IoU$ | $Dice$ | $SDR$ |
| U$^2$-Net (Qin et al., 2020) | 60.8 | 45.2 | 60.6 | 7.85 | 60.2 | 44.8 | 59.9 | 7.70 |
| MTU-NeT (Wang et al., 2022) | 69.1 | 56.5 | 69.0 | 8.17 | 68.3 | 55.7 | 68.3 | 7.96 |
| Segmenter (Strudel et al., 2021) | 72.6 | 59.6 | 72.5 | 9.24 | 70.8 | 57.7 | 70.7 | 8.52 |
| SegNet (Badrinarayanan et al., 2017) | 77.5 | 66.9 | 77.5 | 9.55 | 76.1 | 65.3 | 76.2 | 9.43 |
| R-CED (Park & Lee, 2016) | – | – | – | 2.38 | – | – | – | 1.93 |
| Noise2Noise (Kashyap et al., 2021) | – | – | – | 2.40 | – | – | – | 1.96 |
| TS-U-Net (Moliner & Välimäki, 2022) | – | – | – | 2.48 | – | – | – | 1.98 |
| DVAD (Zhang & Li, 2023) | 82.6 | 73.5 | 82.6 | 10.3 | 81.6 | 72.3 | 81.6 | 9.96 |
| PtDeepLab (Li et al., 2023) | **83.4** | **75.9** | **83.4** | 10.5 | **83.1** | **75.4** | **83.0** | 10.4 |
| **AIGD** | – | – | – | **11.5** | – | – | – | **10.8** |

# F    COMPUTATION TIME

We also list the computation time of our AIGD model across three datasets, as shown in Tab. 5. Given our AIGD model has the $T$ steps generation process, the computation time for each audio is increased during the training. Across three datasets, the mean computing time of training per audio is around 2.4 minutes. The major reason is because of the diffusion process and the gradient computation. Our mean inference time of the three datasets is 1.07 seconds per audio. However, compared with the best baseline methods MANNER (Park et al., 2022), FS-CANet (Chen et al., 2022a) and PtDeepLab (Li et al., 2023) of three benchmark datasets respectively, the computation cost of our AIGD model is still in a reasonable range.

Table 5: Computation time of three benchmark datasets (M: minutes, S: seconds).

| Time (per audio) | VoiceBank-DEMAND | | DNS 2020 challenge | | BirdSoundsDenoising | | |
|---|---|---|---|---|---|---|---|
| | Training | Test | Training | Test | Training | Validation | Test |
| MANNER (Park et al., 2022) | 1.04 M | 0.86 S | - | - | - | - | - |
| FS-CANet (Chen et al., 2022a) | - | - | 1.84 M | 0.83 S | - | - | - |
| PtDeepLab (Li et al., 2023) | - | - | - | - | 0.99 M | 0.79 S | 0.82 S |
| **AIGD** | 2.03 M | 1.09 S | 3.12 M | 1.17 S | 1.98 M | 1.02 S | 1.03 S |

# G    EXAMPLE DENOISED AUDIOS

In our supplementary materials, we also attached six samples of denoised audios, including three human and three bird sound denoising results. In each sample, the suffix with "_noise" is the original audio, "_ground_truth" means the ground truth clean audio, and "_AIGD_denoised" means the denoised results from our proposal AIGD model. From these six samples, we can hear that the denoised audios from our AIGD model extremely approach the ground truth audios, which also reveals that our AIGD model is suitable for audio denoising tasks.

