# OpenReview forum: "Audio Image Generation for Denoising"
_ICLR.cc/2024/Conference — ICLR 2024 Conference Withdrawn Submission_

### Official Review · Reviewer_bMBJ · 2023-10-25

**Soundness:** 2 fair
**Presentation:** 1 poor
**Contribution:** 3 good
**Rating:** 3
**Confidence:** 4

**Summary:**

The paper proposed to train an image diffusion model + iSTFT for audio denoising. Both the spectrogram and phases are modeled. The paper proposed a set of loss functions that could improve denoising quality. A number of denoising studies have been conducted on several standard benchmarks.

**Strengths:**

- It is a novel idea to model both the spectrogram and phases. In the literature most time-frequency methods only model the spectrogram, and this paper presents a novel complementary research to these.
- The set of loss functions can be useful for improving denoising quality and could potentially be used in follow-up research.
- The pesq and other evaluation metrics outperform many baseline methods.

**Weaknesses:**

- The writing of this paper is extremely hard to understand, especially when it comes to section 3.4.2. The proposed method just seems to be a special formula of VE-DDPM with $x_0$ prediction.
- The paper fails to connect the proposed method with inverse problems, Schrodinger bridges, and cold diffusion, although there should be discussion from these perspectives given that the paper models $I^T$ based on (15).
- The proposed method is not causal because it denoises a fixed-size stft outputs. Therefore it is unfair to compare to the baseline methods that are causal.
- The paper also did not compare to other strong baselines, such as SN-Net, CleanUNet, Universe, etc. Their denoising performances are similar and some of them are causal. The paper also did not report MOS-SIG/BAK/OVRL which are crutial metrics.
- Finally, the paper looks very rushy, with a number of typos and scrachy layout. For example, "n_fft=1023" in section 4.1, math symbols not printed in the math mode, code-like math equations, etc. I encourage the authors to have a careful check to the presentation of the paper.

**Questions:**

- What is the latency and the real time factor of the denoising model?

---

### Official Review · Reviewer_dehP · 2023-10-30

**Soundness:** 1 poor
**Presentation:** 2 fair
**Contribution:** 2 fair
**Rating:** 3
**Confidence:** 3

**Summary:**

The paper consider the audio denoising problem. The authors adopt an approach based on cleaning up the complex-valued time-frequency images obtained by applying a short-time Fourier transform to the input signal, and resorting to a variant of the denoising diffusion probabilistic model. The authors note that their approach removes the forward process in the diffusion models (which is responsible for adding noise to a clean observation), in order to denoise the noisy observations. They also employ an image quality term in their cost function to promote a realistic image as the denoised output.

**Strengths:**

The paper aims to modify a relatively recent but promising framework to tackle a practical problem of interest.

**Weaknesses:**

The paper is not always clear. Specifically, Section 3.4.2, which constitutes the main description could use a rewrite. It's not obvious what is being assumed, what is being derived. I'm also not sure if the authors claim that they completely avoid the forward process (where noise gets added) holds. It's true that the noise addition does not completely destroy the signal like in diffusion models, because the signal is not multiplied with a gain less than unity, but noise is still added to the observation, following a particular recipe.

Considering the main loss function in eqn (27) (which doesn't include the image quality term), it appears that a short explanation of the overall idea is to come up with a neural network that can handle different amounts of noise added to a clean signal. I'm not sure if this is entirely a new idea.

Finally, I missed some details on how the method would be applied. Do we need to know the amount of noise in the signal, in order to apply the method?

**Questions:**

Here are some questions, some of which are minor:
- Eqn 14 : Is the last term with $p$ missing a $\theta$ as subscript?
- Eqn 15, minor note on notation : Why not use $\mu_t$, $\sigma_t$ instead of $\mu_{rt}$, $\sigma_{rt}$? The subscript $r$ appears like a multiplicative factor.
- Eqn 15 : How do you generate $\mu_{rt}$, $\sigma_{rt}$? Please include a description here.
- Eqn 16 : Is this your main departure from the regular "diffusion denoising probabilistic model"? You're keeping the signal energy constant throughout the forward process. If so, please emphasize this verbally here.
- Eqn 24 : Isn't $I^T$ fixed to the observed noisy signal? How do you ensure that the noise addition process described here converges to $I^T$? I'm also confused by the statement "We can also assume that" -- why can you assume this? Aren't there constraints that the noise should satisfy?
- Right after Eqn 24, "Proofs can be found in the supplementary material" : What are the statements that you're proving? Please clearly note in the main text, paying attention to what's being assumed etc.
- Eqn 27 : How does this differ from a scheme, where we take a clean signal, add noise of different strengths and then simply denoise? If the index $t$ really playing a crucial role here? Have you done experiments where an incorrect $t$ was input intentionally?
- Algorithm 1, line 8 : Should we have here $\hat{I} = F(I^t, t, \theta)$?
- Section 4.1 : 1000 samples at 48 KHz evaluates to ~20 msec, which is a bit short for audio, but still acceptable. Can you specify the sampling frequencies of the databases you used?
- Section 4.1 : According to the description, the hop-size depends on the length of the clip. Can you give the range of the hop sizes?

---

### Official Review · Reviewer_GkZX · 2023-11-01

**Soundness:** 1 poor
**Presentation:** 1 poor
**Contribution:** 2 fair
**Rating:** 3
**Confidence:** 4

**Summary:**

The paper describes a generative approach for speech enhancement/denoising.  In particular, it uses diffusion based learning for denoising. The paper tries to directly learn the reverse diffusion step – that is the denoising process without the forward step. The training is mostly guided by reconstruction losses and SSIM metrics (used for image quality) measurements. Experimental results are conducted on multiple datasets to show the efficacy of the method.

**Strengths:**

Generative methods for speech enhancement/denoising is an interesting direction, especially offline non-causal denoising can perhaps benefit a lot from such approaches. Diffusion based learning in particular could be a really good direction. While some work has been done in the past, there is a scope to develop better methods and this paper is such an attempt.

**Weaknesses:**

– As mentioned above, diffusion approaches could be an interesting direction but the paper in the current form suffers from several issues.

– The paper is hard to follow. The motivation behind diffusion learning is not clear and what exactly the paper is trying to address is also not clear.

– I am also not sure why the whole process is described as an “image generation”. There is a lot of literature on spectral mapping for denoising – that is predicting the real and imaginary part of STFT for clean speech. This work is doing the same.

– Several of the technical details are hard to follow and unclear. How do you arrive at Eq (15)/16 ? If there is no forward diffusion process then how is the noise addition assumed to be a Markov chain ?

– It’s also not clear how even the generative process is coming into picture. It appears that the model is trained by standard reconstruction losses (in time domain and STFT-domain).

– In DDPM the number of steps as well as the variance scheduling play a critical role. What are their roles and how were they set here.

– The work of Lu et. al. [2022] seems very relevant. While it has been cited, probably much more direct quantitative and qualitative comparisons with it would be good.

– Can you provide examples to listen ?

**Questions:**

Please respond to the points in the weakness section.

---

### Official Review · Reviewer_ZeJc · 2023-11-03

**Soundness:** 3 good
**Presentation:** 2 fair
**Contribution:** 3 good
**Rating:** 5
**Confidence:** 4

**Summary:**

In this paper, the authors propose a new audio image generation model for audio denoising (AIGD). AIGD converts the audio denoising task into an image generation problem, where it estimates the posterior distribution of clean complex images conditioned on noisy complex images. Given any noisy audio signal, AIGD can directly generate a denoised complex image and output a clean audio signal. Extensive experimental results demonstrate that AIGD achieves state-of-the-art performance in speech denoising by generating high-quality frequency domain images.

**Strengths:**

+ The proposed method is technically sound. It borrowed ideas from the diffusion process and proposed a new audio denoising approach.

+  Extensive experimental results demonstrate that AIGD outperforms the compared methods.

**Weaknesses:**

+ The paper claims to perform audio denoising, but it is actually solving speech enhancement, which is a long-standing research problem in the audio signal processing community. However, the authors do not carefully discuss previous and recent progress in speech enhancement. The authors argue that "the aforementioned methods have a similar feature of using clean audio signals as training targets, meaning that their performance will be limited in low signal-to-noise ratio (SNR) scenarios." However, I believe it is standard to use noisy speech audio as input for a speech enhancement model to reduce audio noise in both training and testing.

+ It is common practice to use 2D FT spectrograms as input to deep neural networks for audio processing. Therefore, I find it strange that the authors refer to their method as an "image generation" process for denoising audio using FT maps as input. I do not fully understand the rationale behind this terminology. I think the paper needs a significant rewriting to fix this issue.

+ What is L_R in Eq. (31)? How do the two terms have different effects on denoising?

+ Images and audio spectrograms possess distinct characteristics. What is the rationale behind using SSIM as a measurement for audio?

+ In the supp, the authors should also provide denosing results from other SOTA approaches for comparison.

**Questions:**

Please address questions in Weaknesses.